# Prediction and stratification of longitudinal risk for chronic obstructive pulmonary disease across smoking behaviors

Yixuan He [1,2,3], David C. Qian[4], James A. Diao[5], Michael H. Cho [6], Edwin K. Silverman[3,6,7], Alexander Gusev [8], Arjun K. Manrai[5], Alicia R. Martin [1,2,3,9] ✉ & Chirag J. Patel [5,9] ✉

Smoking is the leading risk factor for chronic obstructive pulmonary disease (COPD) worldwide, yet many people who never smoke develop COPD. We perform a longitudinal analysis of COPD in the UK Biobank to derive and validate the Socioeconomic and Environmental Risk Score which captures additive and cumulative environmental, behavioral, and socioeconomic exposure risks beyond tobacco smoking. The Socioeconomic and Environmental Risk Score is more predictive of COPD than smoking status and pack-years. Individuals in the highest decile of the risk score have a greater risk for incident COPD compared to the remaining population. Never smokers in the highest decile of exposure risk are more likely to develop COPD than previous and current smokers in the lowest decile. In general, the prediction accuracy of the Social and Environmental Risk Score is lower in non-European populations. While smoking status is often considered in screening COPD, our finding highlights the importance of other non-smoking environmental and socioeconomic variables.

Chronic obstructive pulmonary disease (COPD), characterized by persistent obstruction to airflow in and out of the lungs, is the third leading cause of death globally[1]. While tobacco smoking is widely recognized as the single most important risk factor for COPD, it is now well-established that 20%-30% of COPD cases worldwide consist of never smokers, and only 25% of continuous smokers will develop incident COPD[2–5]. This suggests that other risk factors such as non-smoking exposures and genetic markers also play important roles in pathogenesis.

Heritability estimates for COPD typically range between 20–50%[6,7], and several large genome-wide association studies (GWAS) have uncovered significant genetic risk loci[8–10]. Recently, a composite polygenic risk score (PGS) consisting of over 2 million genetic variants across the genome has been demonstrated to predict incident COPD and age of diagnosis better than previously published genetic risk scores[11–14]. However, a large proportion of phenotypic and disease variance is still unexplained and likely attributable to environmental exposures[15].

To date, studies of COPD have primarily focused on the relationships between a single or a small group of environmental or socioeconomic factors without considering the dense correlations of the exposome[16–20]. There does not exist any metric to summarize the cumulative effects of socioeconomic and environmental exposures beyond smoking for COPD. Previously, polyexposure risk scores which

[1]Analytic and Translational Genetics Unit, Massachusetts General Hospital, Boston, MA, USA. [2]Stanley Center for Psychiatric Research, Broad Institute of MIT and Harvard, Cambridge, MA, USA. [3]Department of Medicine, Harvard Medical School, Boston, MA, USA. [4]Department of Radiation Oncology, Winship Cancer Institute of Emory University, Atlanta, GA, USA. [5]Department of Biomedical Informatics, Harvard Medical School, Boston, MA, USA. [6]Channing Division of Network Medicine, Brigham and Women's Hospital, Boston, MA, USA. [7]Division of Pulmonary and Critical Care Medicine, Brigham and Women's Hospital, Boston, MA, USA. [8]Department of Medicine, Dana-Farber Cancer Institute, Boston, MA, USA. [9]These authors contributed equally: Alicia R. Martin, Chirag J. Patel. ✉e-mail: armartin@broadinstitute.org; chirag_patel@hms.harvard.edu

summarize the cumulative risk of many exposures have been constructed for other common diseases such as type 2 diabetes and cardiovascular diseases and have provided more meaningful predictive performance and risk classification than single risk factors[21,22]. We hypothesize that a similar risk score accounting for socioeconomic and environmental factors beyond smoking will also improve COPD prediction and identify individuals with the highest risk of developing COPD.

COPD disproportionately affects individuals in ethnic minority groups—some of the strongest environmental risk factors for COPD, such as tobacco use and occupational exposures to fumes and chemicals, as well as heritability and susceptibility loci differ greatly in prevalence between populations[23-28]. Despite these differences, most studies have focused on individuals of European ancestry. In genetic studies where the reference population has consisted of individuals of European ancestry, the predictive performance of PGS is attenuated in non-European ancestry populations[29,30]. It is unclear whether this will also be true for environmental and socioeconomic factors.

In this study, we constructed and validated the COPD Socioeconomic and Environmental Risk Score (SERS) in a longitudinal cohort analysis that is conditional on smoking behaviors in the UK Biobank (UKB). We sought to determine whether SERS can predict and stratify disease risk across different smoking behaviors, especially among individuals who have never or rarely smoked. We evaluated our score in a held-out set consisting of multiple racial and ethnic groups to determine the generalizability of socioeconomic and environmental risk factors across populations.

## Results

### Baseline characteristics of the study population

A schematic of our study design is shown in Fig. 1. After excluding related individuals with missing information or who had previous/current COPD diagnoses, our study sample consisted of 320,115 individuals (median age 57, with 209,600 females). Of these, 6422 participants had incident COPD over a median follow-up time of 8.09 years (interquartile range, 1.25 years).

### Developing the COPD socioeconomic and environmental risk score (SERS)

We first tested univariate associations in an EXposure wide association study (EXWAS)[31] between 83 factors in the categories of "Socio-demographics", "Lifestyles and environment", "Residential air pollution", and "Residential noise pollution" and COPD incidence. There were 26 factors that were significant ($P < 0.05$) after correcting for multiple hypothesis testing (Supplementary Fig. 1). We used the EXWAS summary statistics to develop the COPD SERS. (Supplementary Data 1, Supplementary Fig. 1). After applying the PXStools algorithm, the final SERS for longitudinal COPD development consisted of 11 exposures: "Type of accommodation lived in", "Own or rent accommodation lived in", "Alcohol drinking status", "Bread type", "Current employment status", "Nitrogen dioxide (2006)", "Types of transport used", "Types of physical activity in past 4 weeks", "Major dietary changes in the past 5 years", "Attendance/disability/mobility allowance", "Time spent watching TV". Since we were interested in developing a SERS that considered factors independent of smoking behaviors, we did not include smoking status or pack-years as an input exposure but instead adjusted for them in our association testing and SERS derivation. In the multivariable model, socioeconomic status and air pollution factors, such as having disability allowance (hazard ratio (HR) = 1.72, 95% CI 1.46–2.03, $P < 0.0001$), renting compared to owning (HR = 1.66, 95% CI 1.41–1.95, $P < 0.0001$), and NO2 levels (HR = 1.01, 95% CI 1.00–1.01, $P = 1.77 \times 10^{-4}$), were most significantly associated with increased risk of COPD (Supplementary Data 2). Consuming white

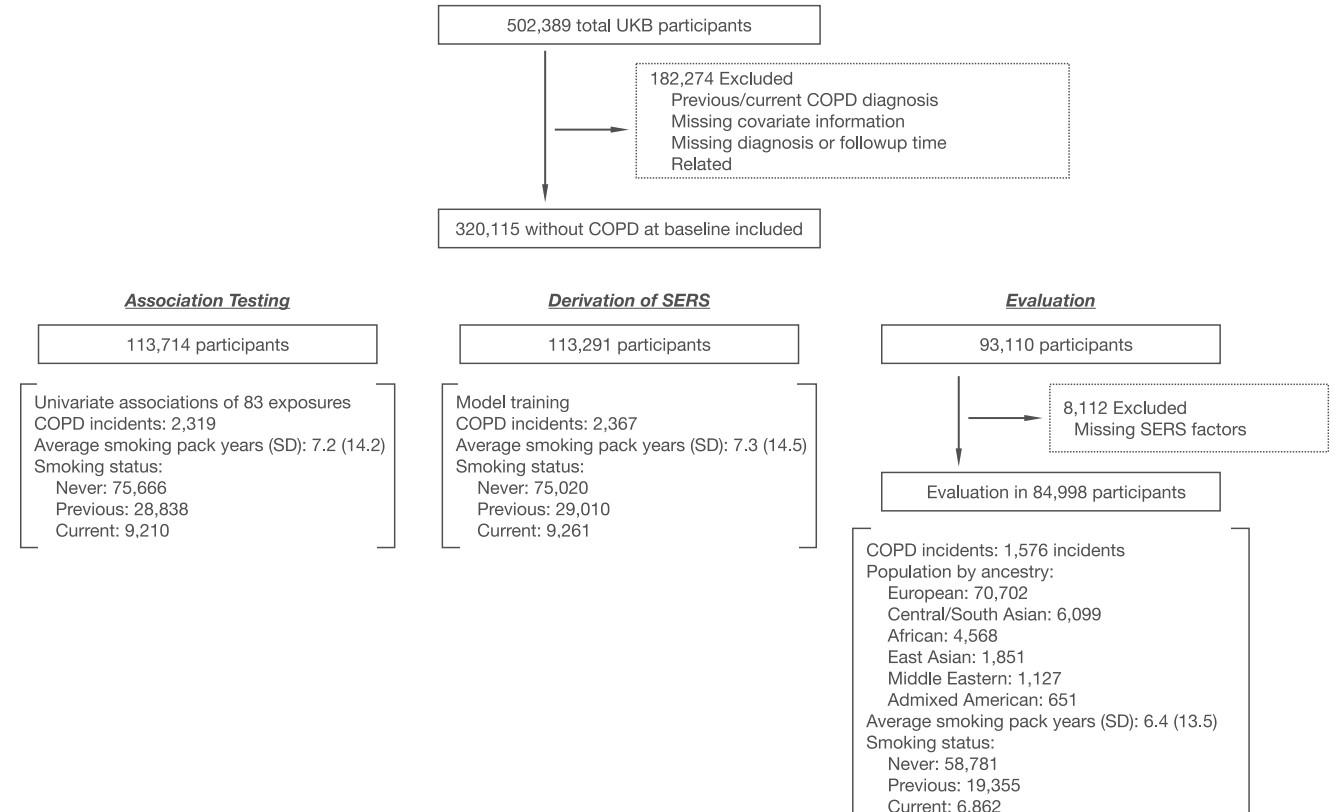

**Fig. 1 | Study design.** COPD Chronic obstructive pulmonary disease, SERS Socioeconomic and environmental risk score, UKB UK Biobank, EXWAS EXposure wide association study.

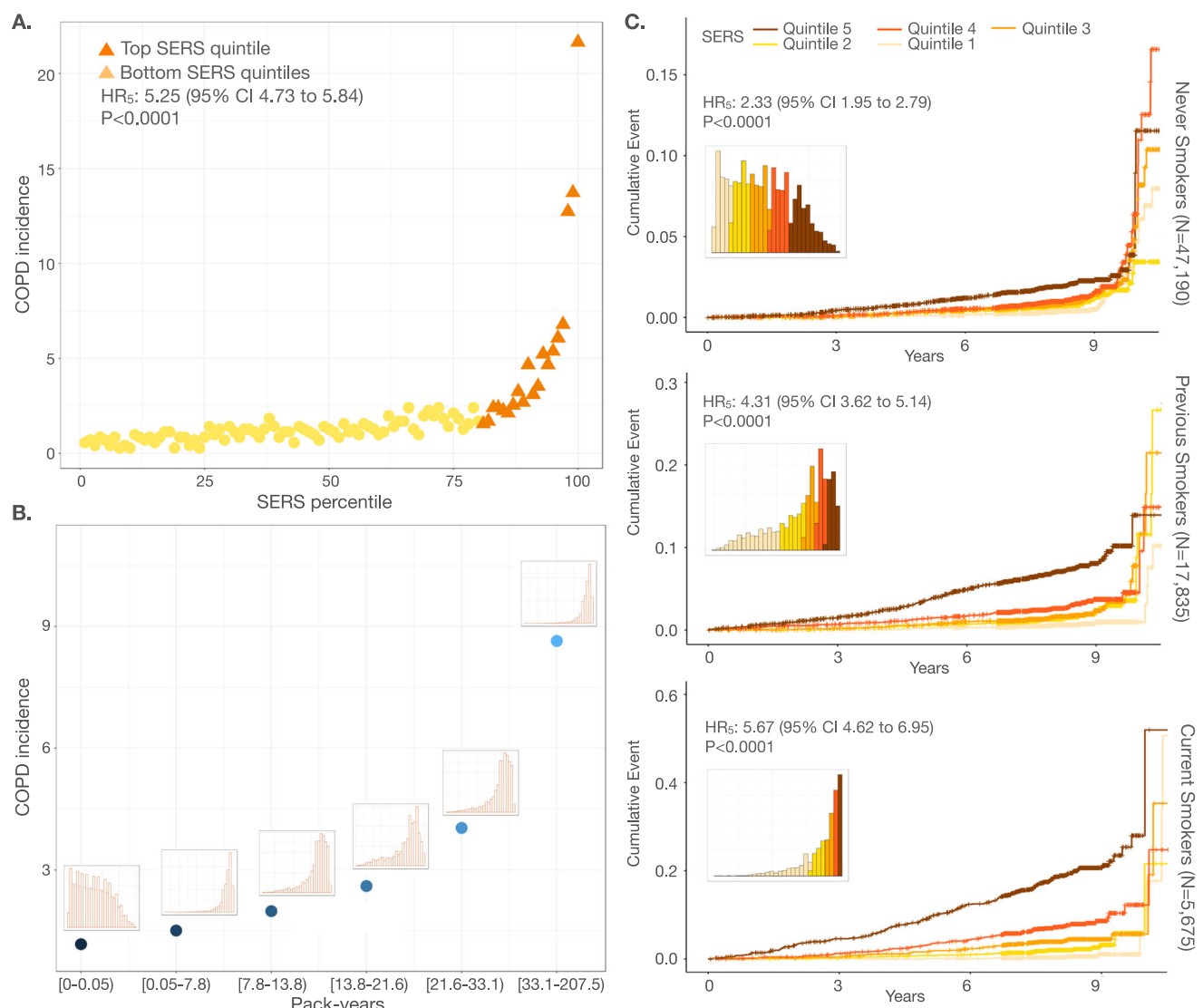

**Fig. 2 | Disease stratification and prediction by SERS across smoking statuses.**
**A** Incidence of COPD in each percentile of the evaluation set ($N = 70{,}702$). The top quintile is colored in dark orange. **B** COPD incidence for each pack-year quintile. The distribution of SERS for individuals in each quintile is shown above each point. **C** Cumulative incidence plots for never smokers ($N = 47{,}190$; top), previous smokers ($N = 17{,}835$; middle), and current smokers ($N = 5675$; bottom) stratified by SERS (orange shades) quintiles. The distribution of SERS for each smoking status is shown in each panel. All displayed $p$ values are two-sided without adjustment for multiple comparisons. Source data are provided as a Source Data file.

bread compared to multigrain (HR = 1.14, 95% CI 1.04–1.26, $P = 8.10 \times 10^{-3}$), being unemployed (HR = 1.49, 95%CI 1.09–2.03, $P = 0.0123$), and being a previous alcohol drinker (HR = 1.23, 95%CI 1.03–1.47, $P = 0.0224$) were also significantly associated with increased risk of COPD. Walking compared to driving a car as the primary source of transportation was significantly associated with decreased risk of COPD (HR = 0.790, 95%CI 0.69–0.91, $P = 7.22 \times 10^{-4}$).

## SERS stratifies the risk of COPD in smoking and non-smoking populations

We first assessed COPD risk stratification by SERS and smoking behaviors (Fig. 2, Supplementary Fig. 2) in the European ancestry population (EUR). We binned individuals by SERS percentiles. Incidence of COPD spanned from 0.28% to 21.64% across SERS percentiles. Compared to the remaining population, individuals in the highest quintile and decile of SERS had HR of 5.25 (95% CI 4.73–5.84, $P < 0.0001$) and 7.24 (95% CI 6.51–8.05, $P < 0.0001$), respectively, for COPD (Fig. 2). The HR of each SERS quintile compared to the first quintile in the EUR evaluation subset can be found in Supplementary Data 3.

SERS predicted incident COPD with a C index of 0.770 (95% CI 0.756–0.784) (Fig. 3), which was significantly higher than both smoking status (C index 0.738, 95% CI 0.725–0.752) and pack-years (C index 0.742, 95% CI 0.727–0.756). In the joint model (C index 0.766 95% CI 0.752–0.780), all three factors were significantly and positively associated with COPD, with pack-years ($P < 0.0001$) being the most significant, followed by SERS ($P < 0.0001$), being a current smoker ($P < 0.0001$), and being a previous smoker compared to a never smoker ($P = 3.02 \times 10^{-2}$). We also assessed the interaction between SERS and smoking behavior. Including interaction terms between SERS with pack-year and smoking status in the joint model did not significantly improve the model performance (C index remain unchanged at 0.770, 95% CI 0.756–0.784), but all interaction terms were significantly associated with COPD incidence ($P < 0.05$).

In the EUR evaluation set, COPD incidence was 548/47,190 (1.44 per 1000 person years) in never smokers, 504/17,835 (3.57 per 1000 person years) in previous smokers, and 373/5675 (8.43 per 1000 person years) in current smokers. Current (HR = 6.10, 95% CI 5.35–6.96, $P < 0.0001$) and previous (HR = 2.56, 95% CI 2.27–2.89, $P < 0.0001$)

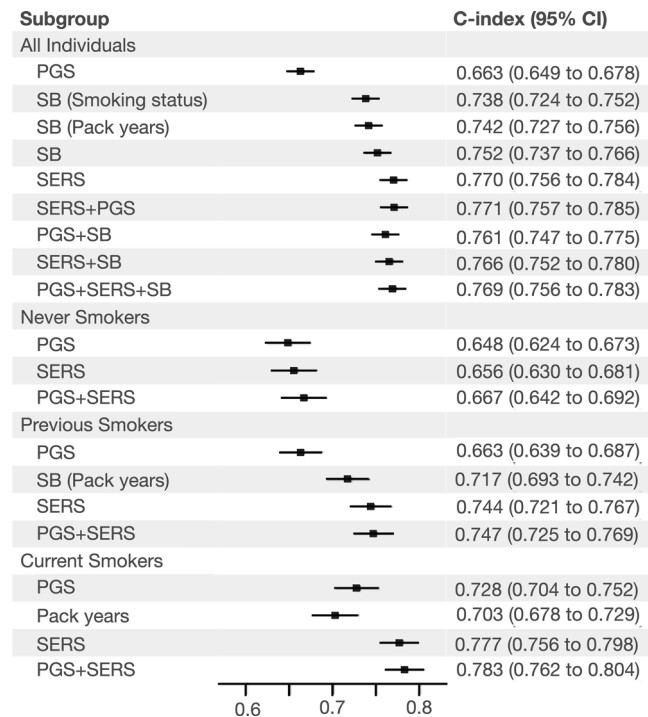

**Fig. 3 | Performance of each prediction model.** The C-indices and 95% confidence intervals for predicting COPD by various models in the entire EUR evaluation set individuals ($N = 70,702$) and across different smoking status subgroups (in never smokers $N = 47,190$, previous smokers $N = 17,835$, and current smokers $N = 5675$). All models include baseline factors sex, age, age$^2$, sex × age, and the first four principal components of genetic ancestry. Data are presented as mean values of the C-indices with 95% confidence intervals. SB smoking behaviors, SERS socioeconomic and environmental risk score, PGS polygenic risk score. Source data are provided as a Source Data file.

smokers had a much higher risk of COPD compared to never smokers. Higher pack-years of smoking were also associated with a higher incidence of COPD and greater risk of COPD with an HR of 1.36 (95% CI 1.34–1.38, $P < 0.0001$) per ten pack-years smoked. SERS was modestly correlated with pack-years ($r^2 = 0.373$, $P < 0.0001$), but SERS was able to stratify COPD risk across smoking behaviors with better granularity. For each smoking status category (never smoker, past smoker, or current smoker), we binned individuals into percentiles by SERS. Regardless of smoking status, COPD incidence increased as SERS increased (Supplementary Fig. 3). Between the highest and lowest SERS deciles, COPD incidence spanned 0.6–2.2% among never smokers, 0.6–9.5% among past smokers, and 0.5–21.0% among current smokers. We also estimated the 10-year cumulative incidence of COPD stratified by SERS (Fig. 2C). In never smokers, previous smokers, and current smokers, individuals in the highest deciles of SERS had an HR of 2.40 (95% CI 1.94 to 2.99, $P < 0.0001$), 5.14 (95% CI 4.13–6.40, $P < 0.0001$) and 5.40 (95% CI 4.48–6.50, $P < 0.0001$), respectively, for developing COPD compared to the remaining population. SERS also outperformed pack years in previous and never smokers—in previous smokers, the C-indices for pack years and SERS were 0.717 (95% CI 0.721–0.767) and 0.744 (95% CI 0.721–0.767), respectively. In current smokers, the C-indices for pack years and SERS were 0.703 (95% CI 0.678–0.729) and 0.777 (95% CI 0.756–0.798), respectively.

Having demonstrated the ability of SERS to predict and stratify risk within smoking status categories, we then investigated if SERS is able to predict COPD risk across different smoking behaviors. Never smokers in the highest SERS decile had an HR of 4.95 (95% CI 1.56–15.69, $P = 6.65 \times 10^{-3}$) and 2.92 (95% CI 1.51–5.61, $P = 1.38 \times 10^{-3}$) compared to current smokers in the bottom decile and quintile of

SERS, respectively. Never smokers in the highest SERS decile also had higher risks for COPD compared to previous smokers in the bottom decile (HR = 4.54, 95% CI 2.39–8.60, $P < 0.0001$) and bottom quintile (HR = 3.49, 95% CI 2.26–5.39, $P < 0.0001$) of SERS. We also found that in individuals who had future COPD incidence, one decile increase in SERS resulted in, on average, 0.26 years shorter time to disease ($P = 7.21 \times 10^{-4}$) (Supplementary Figs. 4, 5).

## Combining genetic, environmental, and socioeconomic factors to predict COPD

To assess the complementarity and additivity of polygenic risk, we computed a composite genome-wide PGS from published weights of 2.5 million SNPS that is predictive of incident COPD and age of onset[11,13,14]. In our study population, we found that the PGS had lower predictive accuracy than smoking behaviors or SERS in the entire evaluation cohort (C index = 0.663, 95% CI 0.649–0.678) as well as within each smoking group (Fig. 2, Supplementary Fig. 6). The composite PGS is able to also stratify risk of COPD. Individuals in the top decile of PGS had an HR of 1.69 (95% CI 1.51–1.89, $P < 0.0001$) compared to the rest of the population (Supplementary Data 4). In never smokers, previous smokers, and current smokers, individuals in the highest deciles of PGS had an HR of 1.74 (95% CI 1.38–2.18, $P < 0.0001$), 1.66 (95% CI 1.25–2.21, $P = 4.56 \times 10^{-4}$) and 1.90 (95% CI 1.51–2.39, $P < 0.0001$), respectively, for developing COPD compared to the remaining population. PGS was also unable to stratify non-smokers who had a higher risk of COPD than previous or current smokers.

To measure gross gene-environment correlation, we estimated the Pearson correlation coefficient between SERS and PGS. Socioeconomic and environmental factors were independent of genetic risks ($P = 0.3$) in the EUR evaluation population. To further investigate genetic and environmental interactions in COPD, we classified individuals into five categories based on whether they were in the top or bottom quintiles of SERS and PGS: high SERS and high PGS, high SERS and low PGS, low SERS and high PGS, low SERS and low PGS, or none of the above. 209/2761 (7.6%) of individuals with both high SERS and PGS were later diagnosed with COPD, while 4/2755 (0.15%) of individuals with both low SERS and PGS were later diagnosed with COPD. Individuals with high SERS and high PGS had an HR of 4.80 (95% CI 4.14–5.56, $P < 0.0001$) for COPD compared to the rest of the population (Supplementary Fig. 7). To investigate how either SERS or PGS may identify risk not implicated by the other score, we compared individuals with high SERS and low PGS, and individuals with low SERS and high PGS. Individuals with high SERS and low PGS had an HR of 4.50 (95% CI 3.08–6.57, $P < 0.0001$) for COPD compared to individuals with low SERS and high PGS, suggesting that low cumulative exposure risk may mediate high cumulative genetic risk. Of all individuals who were later diagnosed with COPD (1,380), there were 435 (31.5%) individuals with only high SERS, 175 (12.7%) with high PGS, and 209 (15.1%) individuals with both high SERS and high PGS (Supplementary Fig. 8).

## Evaluate prediction models in diverse populations

In the testing set, there were 14,296 total non-European ancestry individuals: 6099 individuals of Central/South Asian (CSA) ancestry, 4568 of African (AFR) ancestry, 1851 of East Asian (EAS) ancestry, 1127 of Middle Eastern (MID) ancestry, and 651 of Admixed American (AMR) ancestry. COPD incidence was much lower in these populations. There were 9 (1.78%) AMR incident cases, 38 (1.09 per 1000 person years) AFR cases, 61 (1.30 per 1000 person year) CSA cases, 19 (1.30 per 1000 person years) EAS cases, 20 (2.27 per 1000 person years) MID cases, and, for a total of 147 (1.34 per 1000 person year) incidents cases in non-European populations.

We calculated SERS for non-European ancestry populations using weights derived from the EUR reference population, which predicted COPD risk with a C index of 0.739 (95% CI 0.695 to 0.760) for SERS and PGS, respectively. We then randomly subsampled 1500 individuals

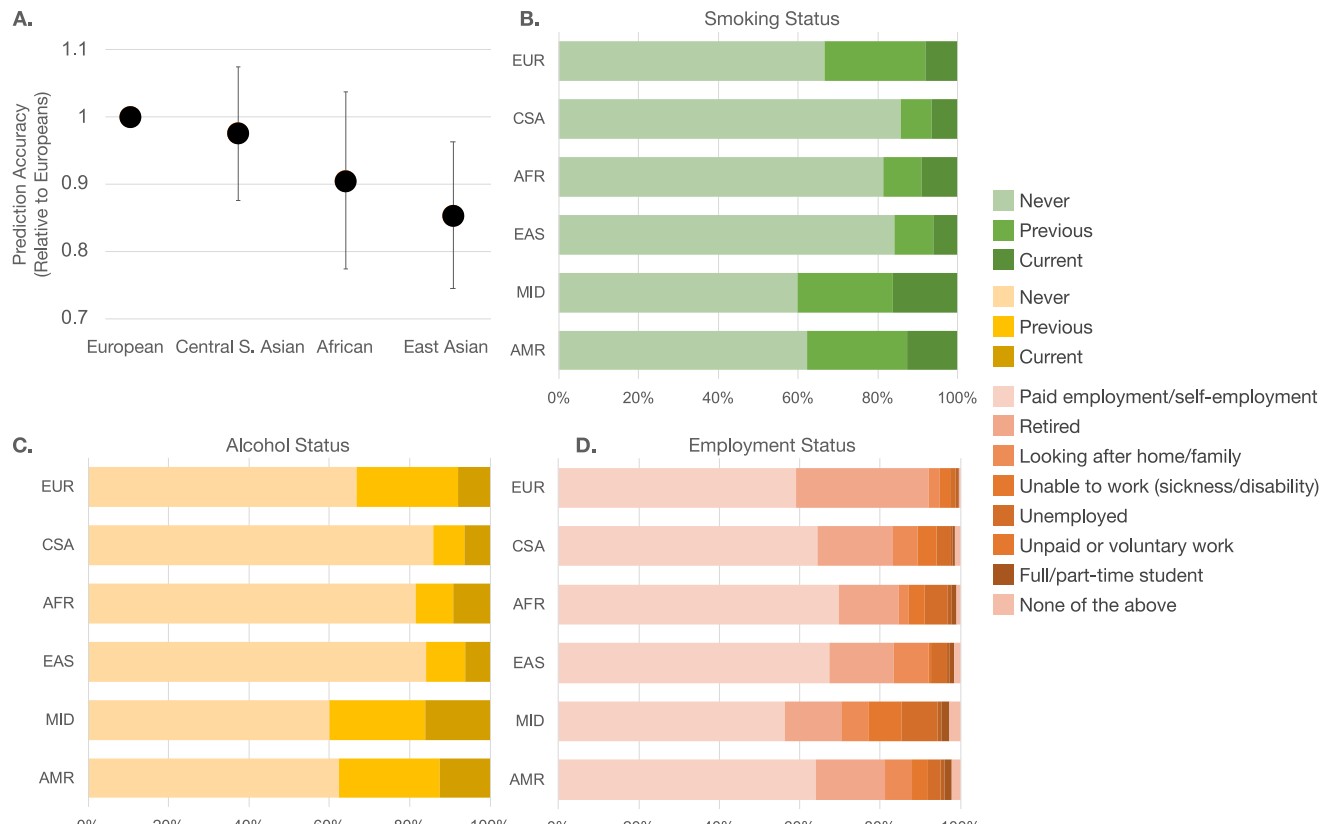

**Fig. 4 | Prediction of COPD across ancestry groups by SERS. A** Prediction accuracy of 1500 individuals randomly subsampled from each of the three largest non-European ancestry subgroups relative to European ancestry individuals with standard error bars. Data are presented as mean values with 95% confidence intervals. Distribution of (**B**) smoking status (left to right: Never, Previous, Current), (**C**) alcohol status (left to right: Never, Previous, Current), and (**D**) employment (left to right: In paid employment or self-employment, Retired, Looking after home and/or family, Unable to work because of sickness or disability, Unemployed, Doing unpaid or voluntary work, Full or part-time student, None of the above) across all ancestry groups. Source data are provided as a Source Data file.

from each of the four largest ancestry groups: EUR, CSA, AFR, and EAS each for subsequent analyses. SERS had worse prediction in all three non-European population subgroups compared to the European ancestry subgroup. (Fig. 4a). We investigated the distribution of smoking status (Fig. 4b) and two SERS exposures, qualifications, and accommodations in the entire evaluation population (Fig. 4c–d). The CSA population had the largest proportion (85.8%) of never smokers, followed by AFR (81.5%), EAS (84.1%), and EUR (66.7%). In the UKB population, CSA, AFR, and EAS ancestry populations consistently had the highest proportion of never-alcohol drinkers and being in paid employment. The absolute values can be found in Supplementary Data 5.

## Discussion

We developed a SERS associated with time to COPD that is trained and evaluated on socioeconomic, environmental, and behavioral variables beyond smoking. The score is able to identify individuals with the highest risk of disease across smoking statuses.

Cigarette smoking is well-established as the greatest risk factor for COPD. However, a striking proportion of 20%–30% of COPD cases worldwide consists of never smokers[2,3]. Previous studies have investigated the associations between a small set of exposures such as air pollution and occupational exposures (e.g., gas and chemicals)[18–20] and COPD. However, individuals are simultaneously exposed to an extensive breadth of other factors. Thus, considering broader categories of environmental exposures to assess risk for COPD[21,32] may be useful to screen populations beyond smoking.

In this study, we used a data-driven approach to build the COPD SERS, which includes 11 indicators of alcohol consumption, air pollution, diet, employment, household information, physical activity, and sociodemographics information, that captures holistic socioeconomic and environmental risk beyond smoking status. Our approach re-highlighted previously reported associations between COPD risk and socioeconomic and environmental factors such as air pollution[33,34], alcohol consumption[35,36], physical activity[37,38], and employment status[39]. Other risk factors that showed strong associations with COPD in our univariate XWAS procedure were not included in our final SERS model as our approach implements a shrinkage and selection procedure that favors interpretability and independence over complexity. For example, while several correlated measures of air pollution such as PM2.5, PM10, NO, and $NO_2$ are established risk factors for COPD, previous studies have shown $NO_2$ to have the highest risk for COPD[34]. In our data-driven procedure, PM2.5, NO, and $NO_2$ were all significantly associated with COPD incidents in univariate association, but only $NO_2$ was retained for the final multivariable model.

SERS achieved marginally better predictive ability for COPD than smoking status and pack years in the total population. However, within smoking status subgroups, SERS outperformed pack years. SERS was also able to identify never smokers with higher risk of COPD. While risk models for COPD have been proposed[40], none, to our knowledge, have used longitudinal biobank-level data to assess the independent risk of smoking, socioeconomic, and environmental factors on incident COPD. For instance, Chen at al.[41] modeled FEV1 and FVC decline using data from four thousand participants in the Framingham Offspring Cohort. Their model is composed of 20 factors including pack years, laboratory blood measurements (e.g, white blood cell count), and diseases and symptoms and achieved C-statistics between 0.86 and

0.87. Guo et al.[42] developed a COPD prediction model consisting of early life factors, genetic polymorphisms, and smoking history using cross-sectional data from roughly 700 Chinese individuals. Future modeling approaches should test how complementary, or not, phenotypic or blood measures are with SERS-like factors in prediction of COPD.

We compared SERS against a composite genome-wide PGS from published weights of 2.5 million SNPS that has been previously demonstrated to be the most predictive genetic risk score for incidence and age of onset of COPD to date[11,13,14]. In our study population, we found that the PGS had significantly lower predictive accuracy than smoking behaviors or SERS in the entire evaluation cohort as well as within each smoking group. PGS was also unable to identify never smokers with an elevated risk of COPD compared to individuals who were previous or current smokers. SERS and PGS were not significantly correlated with each other, and participants with both elevated SERS and PGS had a much greater risk for disease compared to those with only one elevated score. We also investigated whether SERS and PGS may rescue the risk conferred by the other score. Such phenomena would be expected under a liability threshold model, in which genetic and environmental effects combine to determine an individual's total disease liability[43]. Individuals with high SERS and low PGS had an HR of 4.50 for COPD compared to individuals with low SERS and high PGS, suggesting that the effects of genetic risk on COPD depend on the risk conferred by environmental factors. These results are supported by COPD genetic loci related to nicotinic acetylcholine receptors and smoking-related behaviors (e.g., *CHRNA3* and *AGPHD1*)[44].

COPD is a disease that has well-documented disparities between groups worldwide. While PGS has been shown to be far more accurate in European than non-European ancestry groups[29], it was unclear if a similar trend would hold for socioeconomic and environmental factors. Unlike PGS, which decays in accuracy from the study populations as a function of ancestry and genetic distance, SERS performance is driven by cultural, racial, socioenvironmental, and other phenomena. We investigated the generalizability of SERS for predicting COPD risk in non-European ancestry populations by evaluating the performance of SERS in several subsets of non-European ancestry populations in the UKB. The prediction accuracy was consistently lower in the non-European ancestry populations compared to the European evaluation set. In our study population, there were differences in the makeup of some of the most important factors of SERS. For example, CSA, AFR, and EAS ancestry populations had a much smaller proportion of alcohol drinkers and a much higher proportion of being in paid employment compared to the EUR ancestry population. We recognize, however, that the smaller sample size of non-European individuals in the UKB results in lower power and confidence in our conclusions, and that certain exposure factors such as bread type, may not be familiar and generalizable to every population. Future studies should strive to validate our results in larger datasets with more diverse characteristics and ancestry backgrounds, such as in the All of Us Research Program[45]. Furthermore, participants included in the UKB may not be representative of the general population as they tend to be older and are prone to healthy volunteer selection bias[46]. We recognize that an inherent challenge with generalizing results from case-control studies is that ascertainment induces positive correlations between genetic and environmental effects where none may exist in the unascertained population[47]. While SERS has the potential to guide surveillance and facilitate personalized care for COPD, our results should be replicated and carefully validated in datasets and randomized experiments with longer-term follow-up that collect similar environmental and behavioral instruments.

There are notable limitations to using biobank-derived samples with linked diagnosis information to study the association between exposures and COPD. First, there is substantial underdiagnosis and misdiagnosis of COPD in clinical care[48]. While our study used spirometry as additional diagnostic criteria, only a fraction of patients had repeated follow-up measurements. Participants with COPD diagnosed from hospital admission records may also consist of more severe cases. However, these participants are crucial to study as severe diseases require more resources and experience greater morbidity and mortality. The model's predictive performance for milder COPD symptoms may be more uncertain. It is also important to be aware that EHR documents the time of diagnosis but not necessarily the time of disease onset. While easy to measure, self-reported exposures may be prone to measurement error and recall bias[49]. In our study, we assumed that these errors occur at random across all variables considered in the SERS. We excluded exposure variables with >10% missingness, but future studies with more completeness of variables, such as by imputing missing exposure information, would be valuable. One of the most significant challenges in single cohort observational studies such as the UKB is deducing the direction of causality or potential confounding exposure variables. By excluding individuals who at baseline had a past or current diagnosis of COPD, we are more confident that the socioeconomic and environmental risk factors in our study conferred risk for COPD. However, it is possible that some exposures in the SERS (e.g., response to major dietary changes in the past 5 years) may be explained by other comorbidities. Further studies using causal inference approaches such as Mendelian randomization can better inform the directions of effects between exposures and disease[50,51]. This is particularly relevant when considering scenarios where diagnostic biases are likely, such as whether smokers are more likely to be diagnosed with COPD regardless of underlying genetic liability. Understanding genetic associations is especially important for considering interventions as it can inform biological pathways and mechanisms of COPD. In our study, our model gives the most generous estimate for the PGS as the original GWAS used in developing the PGS contained some overlapping samples from the UKB. Despite this, our estimates of prediction are consistent with previous reports of AUC for predicting COPD[11]. While the composite PGS we used is based on multi-trait analysis of quantitative spirometry GWAS, it has been demonstrated to be more predictive of COPD cases and time to diagnosis than other existing PGS. It is, however, unclear how much could be improved by considering a multi-trait analysis of genetically correlated traits, such as GWAS of COPD, asthma, and other phenotypes relevant to lung function[52].

Until recently, studying the cumulative effects of environmental exposures has not been possible on a large scale. With the rise in population-level "biobanks" and high-dimensional epidemiological cohort 'omics data, there are new opportunities to systematically consider a greater range of non-genetic factors. Leveraging the data available from the UKB, we constructed and validated the first COPD risk score that summarizes the risk conferred by a broad set of socioeconomic factors and non-smoking environmental exposures.

## Methods

### Ethics statement

Data for this study were obtained from the UKB Resource under Application Number 22881. All participants from the UKB provided written informed consent for anonymized data to be used for research and publication. This study was conducted in accordance with the criteria set by the Declaration of Helsinki. The Harvard Internal Review Board deemed our investigation as Not Human Research (IRB16-2145).

The UKB is a large observational study of over half a million participants between 40–69 years of age at the time of recruitment between 2006 and 2010[53]. In our analysis, we excluded individuals who had a COPD diagnosis prior to the time of assessment, had missing diagnosis or follow-up time, were related, or had missing covariates. There were 320,115 individuals remaining. We used ancestry assignments from the Pan-UK Biobank (PanUKBB) project[54], which was downloaded through the UKB portal as Return 2442. Based on the

ancestry assignments, we identified 358,627 Europeans (EUR), 8284 Central South Asians (CSA), 6446 Africans (AFR), 2641 East Asians (EAS), 1578 Middle Easterners, and 970 AMR. In brief, PanUKBB conducted a pan-ancestry analysis of the UKB by comparing the genome of UKB participants against two large diverse global datasets, the 1000 Genome Project and the Human Genome Diversity Project, and assigned ancestry based on genetic similarity. The European participants were randomly divided into three subgroups, in a roughly 3:3:2: ratio (association testing $N = 113,714$, derivation of SERS $N = 113,291$, evaluation $N = 93,110$). The evaluation subset, used for assessing the performances of the risk scores, also contained all non-European individuals. We used the entire association testing subgroup to conduct the initial exposure-wide association study (XWAS) on COPD. SERS was calculated for 84,998 individuals in the evaluation subgroup who had complete exposure responses for the final SERS factors.

## Phenotype ascertainment

We classified COPD based on a combination of linked hospital admission records for International Classification of Disease (ICD) 9 codes of 490, 491, 492, 494, 496 ICD-10 codes of J41.X, J43.X, J44.X, J98.2, J98.3, having a forced expiratory volume (FEV1)/forced vital capacity (FVC) ratio of <0.70, or having self-reported COPD at baseline or in repeated follow-up interviews. We used the earliest recorded time of linked admission records, self-report, or lung function assessment of COPD as the time of event in our analysis. In the full population, we identified 8,632 individuals diagnosed with COPD by self-report, 14,677 by ICD10 code, and 30 by ICD9 code. We excluded individuals who had a COPD diagnosis at the time of the first assessment. There were also 50,599 additional individuals who had an FEV1/FVC ratio of <0.70 at the time of assessment who were excluded from our study analysis.

## Socioeconomic and environmental risk score derivation and validation

SERS captures the cumulative impact of socioeconomic, environmental, and behavioral exposure risks. Individuals receive a score based on the weighted sum of many common non-genetic factors to which they may be exposed. Weights are determined by the strength of corresponding associations with the outcome of interest.

We derived SERS using methods from the R package PXStools[21,22]. We first conducted an EXWAS for incident COPD in the derivation subgroup[31,32]. We then iterated through a LASSO-based stepwise selection procedure to identify independent features associated with longitudinal COPD development in the testing subgroup. We calculated the final SERS for the evaluation subgroup by taking the weighted sum of the exposure variables. In each step, we adjusted for sex, age, age$^2$, age × sex, the first four principal components of genetic ancestry, smoking status, and pack-years.

The initial set of exposure variables we included were indicators of physiological state, environmental exposure, and self-reported behavior collected during the first assessment visit period (2006–2010). We wanted to construct a risk score separate of smoking effects, thus we did not consider any exposures in the "Smoking" category. Altogether, we started with 102 unique variables in total. Among these, we only considered variables that had less than 10% missingness, resulting in 83 variables for our pipeline. We processed our exposure data using the PHESANT software tool[55]. Variables belonged to four data types: continuous, ordered categorical, unordered categorical, and binary (Supplementary Data 1). We excluded responses of "Prefer not to answer" and "Do not know". For unordered categorical variables, the response with the largest number of participants was selected as the reference group. Individuals that responded "never smoked" were assigned a pack-year of zero. The final risk score contained eleven independent exposure factors.

## Polygenic score derivation

We reconstructed a composite PGS for lung function and COPD consisting of roughly 2.5 million genetic variants[11]. In summary, Moll et al. derived the most comprehensive and accurate composite PGSs for COPD by training a logistic regression model based on two separate PGSs for forced expiratory volume in 1 s (FEV1) and FEV1/forced vital capacity (FVC) based on results from the GWAS of lung function from UKB and SpiroMeta. The composite PGS consisting of roughly 2.5 million SNPs identified individuals with elevated risk for moderate-to-severe COPD, emphysema subtypes associated with cigarette smoking, and radiographic patterns of reduced lung growth. The PGS has also been demonstrated to be associated with incident COPD and age of onset in large population-based cohorts[13,14].

## Statistical analysis

We conducted all analyses in R version 3.5.1. We placed individuals into bins by their PGS and SERS percentiles and calculated the prevalence of COPD within each bin as well as the hazard ratios for COPD in the top bins of PGS and SERS compared to the remaining individuals. We performed multivariable Cox proportional hazards regressions of COPD on combinations of risk scores and covariates to obtain C-indices for each model. The base model contained only the covariates sex, age, age$^2$, age × sex, and the first four principal components of genetic ancestry. The hazard ratios for membership in certain subgroups versus another subgroup were calculated by fitting a Cox regression model with a binary indicator variable. To measure the gross gene-environment correlation, we estimated the Pearson correlation coefficient between PGS and SERS. The standard errors of relative prediction accuracy were calculated by taking the absolute standard error multiplied by relative accuracy. To adjust for multiple tests, we used the "p.adjust" function of the base stats R package for Benjamini-Hochberg False Discovery Rate adjustment.

## Reporting summary

Further information on research design is available in the Nature Portfolio Reporting Summary linked to this article.

## Data availability

UK Biobank data are available by application via https://www.ukbiobank.ac.uk/. Individual-level genotype data from the UK Bio-banka are available under restricted access to preserve participant privacy. Access can be obtained by researchers through the UK Biobank Data Analysis Platform. The data for this project were accessed through approved protocol 22881. Weights for the composite polygenic risk score were downloaded from http://www.copdconsortium.org/polygenic risk-score. Source data are provided with this paper.

## Code availability

No custom code was developed for this project. To derive SERS, we used the previously developed PXStools software download from https://github.com/yixuanh/PXStools. To calculate PGS, we used PLINK1.90 downloaded from https://www.cog-genomics.org/plink/.

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

## Acknowledgements

This work was supported by the Bioinformatics and Integrative Genomics training grant from the National Institutes of Health NHGRI under award number T32HG002295, the Precision and Genomic Medicine training grant from the National Human Genome Research Institute NHGRI under award number T32HG010464, the National Institutes of Health NIEHS under award number R01ES032470, National Institute of Allergy and Infectious Disease NIAID under award number R01AI12725003, the National Science Foundation Graduate Research Fellowship under award number DGE1745303 (to Y.H.), and the UK Biobank Early-Career Researcher Award (to Y.H.). We are grateful for the volunteers who participated in the UK Biobank.

## Author contributions

Y.H., A.R.M., and C.J.P. designed the study. Y.H. processed, analyzed, and conducted statistical analysis on the data. Y.H., D.C.Q., J.A.D., M.H.C., and E.K.S. interpreted the data. A.R.M. and C.J.P. obtained funding and provided joint supervision. Y.H., D.C.Q., J.A.D., M.H.C, E.K.S., A.K.M., A.G., A.R.M., and C.J.P. all provided critical feedback and revisions for the manuscript.

## Competing interests

E.K.S. has received grant and travel support from GlaxoSmithKline. M.H.C. has received grant support from GSK, consulting fees from Genentech and AstraZeneca, and speaking fees from Illumina. All other authors declare no competing interests.
