## [Peer Review File · Nature Communications]

Prediction and stratification of longitudinal risk for chronic obstructive pulmonary disease across smoking behaviorsReviewer #1 (Remarks to the Author):

The noteworthy results from this study are that the authors have created a risk score (SERS) that moves beyond smoking (as the major risk) for time to COPD onset trained and evaluated on socioeconomic, environmental, and behaviour variables. The need for such a risk score is that 20%-30% of COPD cases worldwide consist of never smokers (with COPD likely caused by non-smoking exposures and genetic markers). The authors also managed to improve risk scoring within smoking status subgroups, where exposures in SERS allowed for stratification of low- and high-risk individuals. They also found that a composite genome wide polygenic risk score had significantly lower predictive accuracy than smoking behaviours or SERS in their cohort.

This work is of significance as little has been published in terms of a COPD metric based on cumulative effects of socioeconomic and environmental exposures (beyond smoking). We also need to know if these risk scores work for non-European ancestry individuals.

This study used the 1/2m participant UK Biobank and included 320,115 individuals in their analysis.

The interpretability of the multivariable model results could be improved by the inclusion of confidence intervals in the second paragraph (as in para 3). From an epidemiological perspective the P values are less informative.

It is perhaps hard to interpret various groups of exposures - more detail on these variables in the main body of the text (Methods) would be welcome (e.g. was NO2 categorial or a numeric values etc. what unit value etc.)?

In the Results section - there are several paragraphs with a first sentence better placed in the introduction or methods. Short subsections and titles might be more appropriate and improve readability.

In the Discussion section, the third paragraph appears to repeat results. Provide a brief summary of these results in the first paragraph of the discussion.

There is some commentary in the discussion provided on the performance of the model. The authors should provide details and make comparison with the current prediction models in the discussion section: e.g. <https://www.nature.com/articles/s41533-022-00280-0>

Minors: Results section page para 3 line 4 - delete 'i' and line 6 '5.2 5'.
Discussion section Missing fullstop para 2.

Reviewer #2 (Remarks to the Author):

Authors developed the SERS to predict COPD, which was tested with stratification of smoking status and with combination of a composite genome-wide polygenic risk score (PGS). It is very noteworthy results, which can emphasize socioeconomic and environmental factors for COPD development.

However, the review and discussion for socioeconomic and environmental factors are necessary. In particular, the factors involved for SERS should be discussed for how these factors are associated with COPD development. etc, not PM10 but NO2 became the factor to predict COPD. Bread type might be an unfamiliar factor to Asian for generalization.

SERS is divided as quintile for COPD prediction. Could you make an each score of factors composed of SERS and set a cut-off total score of SERS for COPD development? , This might be much easier and accessible for clinicians to use in the clinic.

Reviewer #3 (Remarks to the Author):

In this large cohort study, the authors performed longitudinal analysis of COPD in the UK Biobank to develop the Socioeconomic and Environmental Risk Score (SERS) which captures additive and cumulative environmental, behavioral, and socioeconomic exposure risks beyond tobacco smoking. They reported that in addition to genetic factors, socioeconomic and environmental factors beyond smoking can predict and stratify COPD risk for both non- and smoking individuals. The findings of this study might be useful in predicting the risk of COPD. However, the findings are not new and there are several methodological concerns that warrant attention. These are outlined in the comments below.

1. Quite a few prediction models for COPD have been studied/published before (e.g., PMID: 19720809; 31542453). Please compare your results with existing literature and highlight the novelty of the current study.

2. The authors claimed that the prediction accuracy of SERS was lower in the non-European populations compared to the European evaluation set. However, it should be noted that only a very small proportion of participants were non-European, which may explain the finding.

3. "We classified COPD based on a combination of linked hospital admission records for International Classification of Disease (ICD) 9 codes of 490, 491, 492, 494, 496 ICD-10 codes of J41.X, J43.X, J44.X, J98.2, J98.3, having a forced expiratory volume (FEV1)/forced vital capacity (FVC) ratio of < 0.70, or having self-reported COPD in an interview".

To my knowledge, lung function test and interview were only conducted at baseline. The outcome of COPD events was thus mainly determined based on hospital admission records, this could be error-prone in a biobank setting. The majority of COPD patients would not go to hospital, thus the identified COPD cases were mainly severe patients. The authors need to discuss the use of medical records review as a limitation and discuss the specific problems related to using medical records as an outcome.

One possible solution is to leverage additional information (e.g medication) and further curate the data.

4. What's the incident rate of COPD in the follow-up time? Is it comparable with the previous literature?

5. People with more resources are likely to be diagnosed COPD earlier in life, whereas those with fewer resources are more likely to have a delayed diagnosis. One would expect that this would bias estimates.

6. The performance of SERS (C index = 0.770, 95% CI 0.756 to 0.784) is very close to smoking status (C index = 0.738, 95% CI 0.724 to 0.752), pack-years (C index = 0.742, 95% CI 0.727 to 0.756). This implies that smoking is the most important single predictor of COPD. It would be interesting to see the performance of the combination of SERS and smoking.

7. How the time of COPD onset was determined, how accurate is it? Please provide more details.

8. The authors should be aware that Selection bias issues are very important in UK Biobank.

We thank the editor and the reviewers for taking the time to review our manuscript. We have responded in the red color to the reviewer's comments below:

Reviewer #1 (Remarks to the Author):

The noteworthy results from this study are that the authors have created a risk score (SERS) that moves beyond smoking (as the major risk) for time to COPD onset trained and evaluated on socioeconomic, environmental, and behaviour variables. The need for such a risk score is that 20%-30% of COPD cases worldwide consist of never smokers (with COPD likely caused by non-smoking exposures and genetic markers). The authors also managed to improve risk scoring within smoking status subgroups, where exposures in SERS allowed for stratification of low- and high-risk individuals. They also found that a composite genome wide polygenic risk score had significantly lower predictive accuracy than smoking behaviours or SERS in their cohort.

This work is of significance as little has been published in terms of a COPD metric based on cumulative effects of socioeconomic and environmental exposures (beyond smoking). We also need to know if these risk scores work for non-European ancestry individuals.

This study used the 1/2m participant UK Biobank and included 320,115 individuals in their analysis.

We thank the reviewer for their thoughtful comments and finding that the work is novel and of significance.

The interpretability of the multivariable model results could be improved by the inclusion of confidence intervals in the second paragraph (as in para 3). From an epidemiological perspective the P values are less informative.

We thank the reviewer for their suggestion and have included confidence intervals in the second paragraph of our revised results:

*In the multivariable model, socioeconomic status and air pollution factors, such as having a disability allowance (hazard ratio (HR)=1.721, 95% CI 1.46 to 2.03, $P < 0.0001$), renting compared to owning (HR= 1.66, 95% CI 1.41 to 1.95, $P < 0.0001$), and NO₂ levels (HR=1.01, 95%CI 1.00 to 1.01, $P=1.77\times 10^{-4}$), were most significantly associated with increased risk of COPD (**Supplementary Table 2**). Consuming white bread compared to multigrain (HR=1.14, 95% CI 1.04 to 1.26, $P=8.10\times 10^{-3}$), being unemployed (HR=1.49, 95%CI 1.09 to 2.03, $P=0.0123$), and being a previous alcohol drinker (HR=1.23, 95% CI 1.03 to 1.47, $P=0.0224$) were also significantly associated with increased risk of COPD. Walking compared to driving a car as the primary source of transportation was significantly associated with decreased risk of COPD (HR=0.790, 95% CI 0.69 to 0.91, $P=7.22\times 10^{-4}$).*

It is perhaps hard to interpret various groups of exposures - more detail on these variables in the main body of the text (Methods) would be welcome (e.g. was NO2 categorical or a numeric values etc. what unit value etc.)?

We apologize for the lack of clarity and have included the following additional description in the revised methods and Supplementary Tables section:

*Variables belonged to four data types: continuous, ordered categorical, unordered categorical, and binary (**Supplementary Table 1**).*

We also included an additional column "Data Class" in Supplementary Table 1 describing the data class of each variable.

In the Results section - there are several paragraphs with a first sentence better placed in the introduction or methods. Short subsections and titles might be more appropriate and improve readability.

We thank the reviewer for their suggestion and have included the following subsections/titles in the revised results section to improve readability:

Baseline characteristics of the study population
Developing the COPD socioeconomic and environmental risk score (SERS)
SERS stratifies the risk of COPD in smoking and non-smoking populations
Combining genetic, environmental, and socioeconomic factors to predict COPD
Evaluate prediction models in diverse populations

In the Discussion section, the third paragraph appears to repeat results. Provide a brief summary of these results in the first paragraph of the discussion.

We agree with the reviewer and have removed repeated results from the revised discussion section.

There is some commentary in the discussion provided on the performance of the model. The authors should provide details and make comparison with the current prediction models in the discussion section: e.g. <https://www.nature.com/articles/s41533-022-00280-0>

We thank the reviewer for their suggestion. The Shah et al. paper references a progression model for predicting 10-year mortality in patients already diagnosed with COPD. In our revised discussion, we have included the following comparison of our model with previously proposed models for predicting future risk of COPD:

While risk models for COPD have been proposed⁴³, none, to our knowledge, have used longitudinal biobank-level data to assess the independent risk of smoking, socioeconomic, and environmental factors on incident COPD. For instance, Chen et al.⁴⁴ modeled proposed a

prediction model for FEV1 and FVC decline using data from four thousand participants in the Framingham Offspring Cohort. Their model is composed of 20 factors including pack years, laboratory blood measurements, and diseases and symptoms. Guo et al.⁴⁵ developed a COPD prediction model consisting of early life factors, genetic polymorphisms, and smoking history that was constructed using cross-sectional data from roughly 700 Chinese individuals.

Minors: Results section page para 3 line 4 - delete 'i' and line 6 '5.2 5'.
Discussion section Missing fullstop para 2.

We thank the reviewer for pointing this out and have corrected it in the revised manuscript.

Reviewer #2 (Remarks to the Author):

Authors developed the SERS to predict COPD, which was tested with stratification of smoking status and with combination of a composite genome-wide polygenic risk score (PGS). It is very noteworthy results, which can emphasize socioeconomic and environmental factors for COPD development.

However, the review and discussion for socioeconomic and environmental factors are necessary. In particular, the factors involved for SERS should be discussed for how these factors are associated with COPD development. etc, not PM10 but NO2 became the factor to predict COPD. Bread type might be an unfamiliar factor to Asian for generalization.

We thank the reviewer for their suggestion and have elaborated on specific environmental factors in our revised discussions:

In this study, we used a data-driven approach to build the COPD SERS, which includes 11 indicators of alcohol, air pollution, diet, employment, household information, physical activity, and sociodemographics information, that captures holistic socioeconomic and environmental risk beyond smoking status. Our approach re-highlighted previously reported associations between COPD risk and socioeconomic and environmental factors such as air pollution^{36,37}, alcohol consumption^{38,39}, physical activity^{40,41}, and employment status⁴². Other risk factors that showed strong associations with COPD in our univariate XWAS procedure were not included in our final SERS model as our approach implements a shrinkage and selection procedure that favors interpretability and independence over complexity. For example, while several correlated measures of air pollution such as PM2.5, PM10, NO and NO2 are established risk factors for COPD, previous studies have shown NO2 to have the highest risk for COPD³⁷. In our data-driven procedure, PM2.5, NO and NO2 were all significantly associated with COPD incidents in univariate association, but only NO2 was retained for the final multivariable model.

SERS is divided as quintile for COPD prediction. Could you make an each score of factors composed of SERS and set a cut-off total score of SERS for COPD development? , This might be much easier and accessible for clinicians to use in the clinic.

We thank the reviewer for their suggestion and agree that SERS has the potential to guide surveillance procedures and facilitate personalized care. However, implementing a clinical risk calculator into care and decision medicine requires careful validation, potentially in a randomized setting. We have included this as future directions in our revised discussion:

While SERS has the potential to guide surveillance and facilitate personalized care for COPD, our results should be replicated and carefully validated in datasets and randomized experiments with longer-term follow up that collect similar environmental and behavioral instruments. In addition to this validation, future work should replicate our results in larger datasets with more diverse characteristics and ancestry backgrounds, such as the All of Us Project.

Reviewer #3 (Remarks to the Author):

In this large cohort study, the authors performed longitudinal analysis of COPD in the UK Biobank to develop the Socioeconomic and Environmental Risk Score (SERS) which captures additive and cumulative environmental, behavioral, and socioeconomic exposure risks beyond tobacco smoking. They reported that in addition to genetic factors, socioeconomic and environmental factors beyond smoking can predict and stratify COPD risk for both non- and smoking individuals. The findings of this study might be useful in predicting the risk of COPD. However, the findings are not new and there are several methodological concerns that warrant attention. These are outlined in the comments below.

1. Quite a few prediction models for COPD have been studied/published before (e.g., PMID: 19720809; 31542453). Please compare your results with existing literature and highlight the novelty of the current study.

We thank the reviewer for their suggestion. The cited risk prediction models are valuable but have different hypotheses, objectives, and use cases than the model we develop here—PMID 19720809 references the COPD Assessment Test, which evaluates the impact of COPD on health status in patients with existing COPD. In other words, it models progression rather than onset, an important objective, but different than ours. In our revised discussions, we have included the following comparison of our model with previously proposed models for prediction future risk of COPD:

While risk models for COPD have been proposed⁴³, none have used longitudinal biobank-level data to assess the independent risk of socioeconomic and environmental factors on incident COPD. For instance, Chen et al.⁴⁴ modeled FEV1 and FVC decline using data from four thousand participants in the Framingham Offspring Cohort. Their model is composed of 20 factors including pack years, laboratory blood measurements, and diseases and symptoms. Guo et al.⁴⁵ developed a COPD prediction model consisting of early life factors, genetic polymorphisms, and smoking history that was constructed using cross-sectional data from roughly 700 Chinese individuals.

2.The authors claimed that the prediction accuracy of SERS was lower in the non-European populations compared to the European evaluation set. However, it should be noted that only a very small proportion of participants were non-European, which may explain the finding.

We agree with the reviewer that there is a very small proportion of the study sample which were non-European. In our analysis, we randomly subsetted 1,500 individuals from each of the four largest ancestry groups for a more fair comparison. In our discussion section, we additionally state:

We recognize, however, that the smaller sample size of non-European individuals in the UKB results in lower power and confidence in our conclusions, and that, certain exposure factors, such as bread type, may not be familiar and generalizable to every population. Future work should replicate our results in larger datasets with more diverse characteristics and ancestry backgrounds, such as in the All of Us Research Program⁴⁸.

3.“We classified COPD based on a combination of linked hospital admission records for International Classification of Disease (ICD) 9 codes of 490, 491, 492, 494, 496 ICD-10 codes of J41.X, J43.X, J44.X, J98.2, J98.3, having a forced expiratory volume (FEV1)/forced vital capacity (FVC) ratio of < 0.70, or having self-reported COPD in an interview”.

To my knowledge, lung function test and interview were only conducted at baseline. The outcome of COPD events was thus mainly determined based on hospital admission records, this could be error-prone in a biobank setting. The majority of COPD patients would not go to hospital, thus the identified COPD cases were mainly severe patients. The authors need to discuss the use of medical records review as a limitation and discuss the specific problems related to using medical records as an outcome.

One possible solution is to leverage additional information (e.g medication) and further curate the data.

We thank the reviewer for their comments and apologize for the confusion. A fraction of the participants had repeated followup lung function tests and interview data, which was also used in our diagnosis criteria. We have revised our methods to read:

We classified COPD based on a combination of linked hospital admission records for International Classification of Disease (ICD) 9 codes of 490, 491, 492, 494, 496 ICD-10 codes of J41.X, J43.X, J44.X, J98.2, J98.3, having a forced expiratory volume (FEV1)/forced vital capacity (FVC) ratio of < 0.70, or having self-reported COPD at baseline or in repeated followup interviews.

We also agree that there are limitations to using medical records and further elaborated in our discussions:

There are notable limitations to using electronic healthcare records (EHR) to study the association between exposures and COPD. First, there is substantial underdiagnosis and misdiagnosis of COPD in clinical care⁵¹. While our study used spirometry as additional

diagnostic criteria, only a fraction of patients had repeated follow-up measurements. Participants with COPD diagnosed from hospital admission records may be more severe cases. However, these participants are crucial to study as severe diseases require more resources and experience greater morbidity and mortality.

Furthermore, based on the experience from the pulmonologists on our team, we concluded that while including medication usage may potentially be useful, there is also substantial overlap between COPD and asthma treatment that would be challenging to tease apart. Furthermore, many subjects with normal spirometry may be prescribed COPD medication by their doctors.

4. What's the incident rate of COPD in the follow-up time? Is it comparable with the previous literature?

We thank the reviewer for raising these questions. In the revised results, we have included the following description of the incidence rate:

Of these, 6,422 participants had incident COPD over a median follow-up time of 8.09 years (interquartile, 1.25 years)

Since we have excluded participants who have COPD at baseline, our study population may not be representative of the general population. There is also significant underdiagnosis of COPD. We have elaborate on this in our revised discussions, as described in response to point #3.

5. People with more resources are likely to be diagnosed COPD earlier in life, whereas those with fewer resources are more likely to have a delayed diagnosis. One would expect that this would bias estimates.

We agree with the reviewer that there may be population bias in the UKB cohort. In our manuscript, we added this in our discussions:

Furthermore, participants included in the UKB may not be representative of the general population as they tend to be older and are prone to healthy volunteer selection bias.

We also agree that individuals with fewer resources may be diagnosed with more severe stages of COPD. In our revised discussions we have included the following:

Participants with COPD diagnosed from hospital admission records may consist of more severe cases. However, these participants are critical to study as severe diseases require more resources and experience greater morbidity and mortality.

6. The performance of SERS (C index = 0.770, 95% CI 0.756 to 0.784) is very close to smoking status (C index = 0.738, 95% CI 0.724 to 0.752), pack-years (C index = 0.742, 95% CI 0.727 to 0.756). This implies that smoking is the most important single predictor of COPD. It would be interest to see the performance of the combination of SERS and smoking.

We agree with the reviewer that smoking is the most important risk factor for COPD. In Figure 3, we compared various combinations of SERS and smoking behaviors (SB). We have also included additional analysis looking at the interaction between SERS and smoking behaviors, which we described in the revised results under subsection “*SERS stratifies the risk of COPD in smoking and non-smoking populations*”, as follows:

SERS predicted incident COPD with a C index of 0.770 (95% CI 0.756 to 0.784) (Figure 3), which was significantly higher than both smoking status (C index 0.738, 95% CI 0.725 to 0.752) and pack-years (C index 0.742, 95% CI 0.727 to 0.756). In the joint model (C index 0.766 95% CI 0.752 to 0.780), all three factors were significantly and positively associated with COPD, with pack-years ($P < 0.0001$) being the most significant, followed by smoking status ($P < 0.0001$), being a current smoker ($P < 0.0001$), and being a previous smoker compared to never smoking ($P=3.02 \times 10^{-2}$). We also assessed the interaction between SERS and smoking behaviors. Including interaction terms between SERS with pack year and smoking status in the joint model did not significantly improve the model performance (C index remained unchanged at 0.770, 95% CI 0.756 to 0.784), but all interaction terms were significantly associated with COPD incidence ($P < 0.05$).

7.How the time of COPD onset was determined, how accurate is it? Please provide more details.

We apologize for the lack of clarity. In our revised methods we have further elaborated:

We used the earliest recorded time of linked admission records, self report, or lung function assessment of COPD as the time of event in our analysis.

We also further clarified that this study does not evaluate the time of COPD onset but rather the time of COPD diagnosis:

It is also important to be aware that EHR documents the time of diagnosis but not necessarily the time of disease onset.

8.The authors should aware that Selection bias issues are very important in UK Biobank.

We agree with the reviewer that selection bias is important to be aware of in the UK Biobank. In our discussion, we acknowledge this as follows:

Furthermore, participants included in the UKB may not be representative of the general population as they tend to be older and are prone to healthy volunteer selection bias⁴⁹.

Reviewer #1 (Remarks to the Author):

I am happy with these responses.

Reviewer #2 (Remarks to the Author):

The authors have addressed my comments on this round. Thank you.

Reviewer #3 (Remarks to the Author):

The authors have made the necessary modification suggestions appropriately. The article was much better in its revised version. However, it should be noted that Participants with COPD diagnosed from hospital admission records may represent more severe cases. The model's predictive performance for milder COPD symptoms is uncertain.

We are extremely grateful to the editor and reviewers for their time to review our manuscript. We have responded in red color to each comment below:

Reviewer #1 (Remarks to the Author):

I am happy with these responses.

We thank the reviewer for their previous comments and suggestions.

Reviewer #2 (Remarks to the Author):

The authors have addressed my comments on this round. Thank you.

We thank the reviewer for their previous comments and suggestions.

Reviewer #3 (Remarks to the Author):

The authors have made the necessary modification suggestions appropriately. The article was much better in its revised version. However, it should be noted that Participants with COPD diagnosed from hospital admission records may represent more severe cases. The model's predictive performance for milder COPD symptoms is uncertain.

We thank the reviewer for their comments and suggestions. We have included the following in the revised version of the Discussions:

Participants with COPD diagnosed from hospital admission records may consist of more severe cases. However, these participants are crucial to study as severe diseases require more resources and experience greater morbidity and mortality. The model's predictive performance for milder COPD symptoms may be more uncertain.